Hypothyroidism and diabetes mellitus – a risky dual gestational endocrinopathy

Tirosh Dan 1
Benshalom-Tirosh Neta 1
Novack Lena 2
Press Fernanda 1
Beer-Weisel Ruthy 1
Wiznitzer Arnon 1
Mazor Moshe 1
Erez Offer 1 erezof@bgu.ac.il
1 Department of Obstetrics & Gynecology, Soroka University Medical Center , Beer Sheva , Israel
2 Department of Epidemiology, School of Medicine, Faculty of Health Sciences, Ben Gurion University of the Negev , Beer Sheva , Israel
Grundy Scott
Electronic publication date: 2013 Mar 19
Publication date: 2013
Volume: 1
Electronic Location ID: e52
Received 2012 Dec 2; Accepted 2013 Feb 28
Copyright: © 2013 Tirosh et al.
Copyright year: 2013
Copyright holder: Tirosh et al.
License: This is an open access article distributed under the terms of the Creative Commons Attribution License, which permits unrestricted use, distribution, and reproduction in any medium, provided the original author and source are credited.
License URL: https://creativecommons.org/licenses/by/3.0/

Keywords: Preterm labor, Cesarean section, Preterm delivery, Preeclampsia, Pregnancy

Funding: The authors declare that they have no competing interests.

==============================
Objectives. Diabetes mellitus (DM) and hypothyroidism are each associated with increased rate of pregnancy complications. However, their combined morbidity during gestation is poorly studied. Therefore, the aims of this study were to determine the prevalence of the combined morbidity of DM & hypothyroidism and whether it is associated with adverse maternal and neonatal outcome.

Study design. This population based retrospective cohort study included 87,213 women who had 232,293 deliveries. All deliveries were divided into the following groups: (1) hypothyroidism & DM (n = 171); (2) hypothyroidism (n = 1502); (3) DM (n = 13,324); and (4) deliveries of women with neither endocrinopathy, who served as a control group (n = 217, 296).

Results. The prevalence of DM & hypothyroidism in our population was 0.17%. In comparisons to the other study groups, women with DM & hypothyroidism had higher rates of infertility (p < 0.001), preeclampsia (p < 0.001), chronic hypertension (p < 0.001), preterm birth (p < 0.001), and cesarean deliveries (p < 0.001). In Generalized Estimating Equations (GEE) model, hypothyroidism & DM was an independent risk factor for cesarean section (OR 3.46; 95% CI 2.53–4.75) and for preeclampsia (OR 1.82; 95%CI 1.16–2.84).

Conclusion. The combination of DM & hypothyroidism is rare, yet it is associated with higher rate of infertility, cesarean sections, preterm deliveries, and hypertensive disorders of pregnancy than the rest of the population. This dual endocrinological combination is an independent risk factor for preeclampsia and cesarean section. These findings suggest that these patients are at risk for perinatal complications and should be followed and delivered as high risk pregnancies.

Introduction

Diabetes mellitus (DM) and hypothyroidism disorders are among the most common endocrinopathies during pregnancy. The prevalence of DM during pregnancy is about 7%, most of them are gestational, of note there is a constant increase of the prevalence of DM due to the gradual increase in pregnant women’s BMI (Body Mass Index) and the epidemic of obesity during recent decades (American Diabetes Association, 2004). Gestational diabetes mellitus (GDM) and especially pregestational DM, are known as risk factors for pregnancy complications, effecting both the mother and the fetus and include among the rest gestational hypertension, cesarean sections, macrosomic fetuses and shoulder dystocia (Casey et al., 1997; Barahona et al., 2005; Langer et al., 2005). These patients also have increased neonatal morbidity including fetal demise, neonatal hypoglycemia, jaundice, polycytemia and hypocalcemia (American Diabetes Association, 2004; Casey et al., 1997; Barahona et al., 2005; Langer et al., 2005). Moreover, maternal diabetes is associated with long term implications on the fetus including increase incidence of future obesity and type II diabetes (American Diabetes Association, 2004).

Hypothyroidism is the second most common endocrinopathy during pregnancy, and its incidence range from 2% to 5%. Autoimmune thyroiditis (also known as Hashimoto’s thyroiditis) and iatrogenic thyroid gland destruction as a therapeutic measure for hyperthyroidism are the most common etiologies for this endocrinopathy in pregnant women (Smallridge & Ladenson, 2001; LeBeau & Mandel, 2006; Nambiar et al., 2011).

Pregnant women with hypothyroidism, experience a higher rate of first trimester abortions (McCanlies et al., 1998; Abalovich et al., 2002; Hallengren et al., 2009), anemia, post-partum hemorrhage, gestational hypertension and placental abruption (Poppe & Glinoer, 2003). Fetuses of pregnant women with uncontrolled hypothyroidism are at a greater risk to develop multinodular goiter and have a higher tendency to be small or large for gestational age (Sahu et al., 2010; Betti et al., 2011). Moreover, in-utero exposure to maternal hypothyroidism increases the risk for miscarriage, intrauterine fetal death and CNS (central nervous system) and developmental disorders as well as mental illnesses and lower than average IQ later on in life (Haddow et al., 1999; Poppe & Glinoer, 2003; Betti et al., 2011). However, continuous monitoring and balancing of thyroid functions decreases the prevalence of most of these complications (Abalovich et al., 2002).

The association between different types of DM and hypothyroidism was previously reported (Perros et al., 1995; Smithson, 1998; Gallas et al., 2002; Van sen Driessche et al., 2009; Papazafiropoulou et al., 2010). Indeed, autoimmune diseases, such as insulin dependent DM, Hashimoto thyroiditis, pernicious anemia and others, are more prevalent among women, show a familial tendency, and may occur concomitantly in a higher rate than their prevalence in the general population (Van sen Driessche et al., 2009). There is inconsistent evidence regarding the association between hypothyroidism and GDM. Some reports found such an association, while others failed to show this connection, but did show an increased risk for later onset diabetes in women who had hypothyroidism during pregnancy, and women with an increased risk for GDM, especially those with familial history of both DM and hypothyroid disorders, also have an increased prevalence of positive antithyroid autoantibodies (Olivieri et al., 2000; Cleary-Goldman et al., 2008; Mannisto et al., 2010). In the literature review we conducted, the information regarding the influence of the combination of these two endocrinopathies on pregnancy and perinatal outcome is lacking.

The aims of this study were to determine the prevalence of the combination of DM and hypothyroidism in pregnant women and to determine whether this dual gestational endocrinopathy is associated with adverse maternal and neonatal outcome.

Material and Methods

The study is a retrospective population based cohort study which included all pregnant women who delivered at the Soroka University Medical Center since 1988 through April 2010 (n = 87,213) who had 232,293 deliveries. All deliveries were divided into the following groups: (1) hypothyroidism & DM (n = 171); (2) hypothyroidism (n = 1502); (3) DM (n = 13,324); and (4) deliveries of women with neither endocrinopathy that served as a control group (n = 217,296). Exclusion criteria included: chromosomal abnormalities or structural defects of the fetus, and multiple pregnancies.

The diagnosis of diabetes or hypothyroidism and the data on demographic characteristics, medical and obstetric history, pregnancy outcomes, including maternal and neonatal morbidity and mortality, were obtained from the computerized database. The diagnoses of the different types of diabetes mellitus and hypothyroidism, used for the statistical analysis, were according to their ICD-9 as recorded in our computerized database. The use of the database was possible as the ‘Soroka’ University Medical Center is a tertiary medical center that exclusively serves the population of the Negev (southern Israel) and all deliveries of the region take place in its labor and delivery suites.

The study complied with the Declaration of Helsinki and ethical approval was obtained by the review board of the Soroka University Medical Center.

Statistical analysis methods

Continuous variables were presented as mean ± SD, median, minimal and maximal values, and compared between groups using t-test and Kruskal-Wallis test, depending on the distribution of each variable. Categorical variables were described as percentages and compared between groups by Chi-Square test. Variables found to be significantly associated with the study outcomes in the univariable analysis (p value < 0.05), were included in a multivariable analysis. At the last stage of modeling, the list of covariates has been reduced to the main exposures at study as well as factors at 0.05 level of significance or variables which we believed provided a necessary adjustment to the main exposures (even if not significant in the current model). We employed an “enter” method at all steps of multivariable analysis.

Adjustment to dependent observations within clusters formed by the same women having more than one delivery within the study period was achieved by Generalized Estimating Equations (GEE) model with binary outcome and logit link function. P-value of 0.05 was considered significant. Analysis was performed on SAS software version 9.0 (Cary, NC, USA).

Results

The prevalence of the combination of DM and hypothyroidism in our study population was 0.17% of all women and 0.07% of all the deliveries at our medical center (Fig. 1).

Figure 1 The prevalence of endocrinopathies in the study population.

The epidemiologic characteristics of the four groups are presented in Table 1. Mean maternal age was significantly lower in the group of healthy women by at least 2 years, than in the other groups. Jewish ethnicity was more prevalent in all groups with a single or a combined endocrinopathy, than in healthy population group (63.7% within the 3 groups of disease vs. 49.2% in healthy population; p < 0.001). Women in the hypothyroidism and DM group had a higher rate of infertility treatments (11.1% vs. 5.2% in the other study groups; p = 0.001), history of preterm birth (15.8% vs. 9.7% in the other study groups; p < 0.007) and a history of two or more spontaneous abortions (8.8% vs. 4.2% in the other study groups; p < 0.003).

Table 1 Demographical and medical background.

Characteristic	Healthy
(n = 217, 296 deliveries, N = 83, 074 women)	Diabetes
(n = 13, 324 deliveries,
N = 9, 771 women)	Hypothyroidism
(n = 1, 502 deliveries,
N = 1, 100 women)	Hypothyroidism + diabetes
(n = 171 deliveries,
N = 146 women)	P-value	
Maternal age, years Mean ± SD	28.3 ± 5.8	32.9 ± 5.9	31 ± 5.2	33.5 ± 5.9	<0.0001	
Jewish origin	49.2%	62.4%	73.8%	77.8%	<0.0001	
Gravidity, 1st pregnancy	20.2%	13.9%	22.2%	19.3%	<0.0001	
2–5th pregnancy	57.5%	49.1%	59.7%	56.1%	<0.0001	
6th pregnancy and more	22.3%	37%	18.1%	24.6%	<0.0001	
Parity, 1st delivery	25.4%	19.7%	29.2%	28.9%	<0.0001	
2–5th delivery	62.4%	56.7%	62%	56.6%	<0.0001	
6th delivery and more	12.3%	23.6%	8.8%	14.5%	<0.0001	
Infertility Treatment	4.9%	9.4%	6.3%	11.1%	<0.0001	
History of fetal mortality	2.4%	3.8%	1.7%	2.3%	<0.0001	
History of preterm birth	9.6%	10.7%	9.1%	15.8%	<0.0001	
History of >2 spontaneous abortions	3.9%	8.1%	6.6%	8.8%	<0.0001	

Table 2 Perinatal characteristics.

Characteristic	Healthy
(n = 217, 296 deliveries, N = 83, 074 women)	Diabetes
(n = 13, 324 deliveries, N = 9, 771 women)	Hypothyroidism
(n = 1, 502 deliveries,
N = 1, 100 women)	Hypothyroidism + diabetes
(n = 171 deliveries,
N = 146 women)	P-value	
Hydramnios	3.1%	11.9%	2.9%	7%	<0.0001	
Oligohydramnios	2.4%	1.9%	3.1%	1.8%	0.0002	
Mild PET	3.2%	5.5%	4.2%	10.5%	<0.0001	
Severe PET	1%	2%	1.2%	3.5%	<0.0001	
Chronic HTN	1.2%	6.6%	2.3%	11.1%	<0.0001	
NPL stage I	1.7%	3.5%	2.4%	2.9%	<0.0001	
NPL stage II	1.6%	1.9%	1.5%	0.6%	0.0159	
PROM	7.8%	6.7%	11.7%	10.5%	<0.0001	
Preterm delivery	7.4%	9.7%	8.6%	14%	<0.0001	
Preterm delivery and PROM	1.2%	1.6%	1.9%	1.8%	0.0001	
NRFHR	1.9%	2.7%	0.3%	0%	<0.0001	
Placenta Previa	0.4%	0.7%	0.5%	0%	<0.0001	
Uterine rupture	0.1%	0.1%	0%	0%	0.7895	
Instrumental delivery	3.2%	2.8%	3.7%	3.5%	0.1043	
Cesarean section	12.3%	27%	23.4%	44.4%	<0.0001	
Malpresentation	4.1%	6.1%	6.2%	7%	<0.0001	
Abruption of placenta	0.7%	0.8%	0.5%	1.8%	0.3074	
Infection of amniotic fluids	0.7%	1.1%	0.7%	1.8%	<0.0001	
Induction	17.5%	35.8%	24.3%	39.2%	<0.0001	
Induction and CS	1.9%	6.5%	3.3%	9.4%	<0.0001	
Induction and Preterm delivery	1.3%	1.6%	2%	4.7%	<0.0001	
Urgent CS	7.1%	14.1%	10.2%	18.7%	<0.0001	
Non urgent CS	5.2%	12.9%	13.2%	25.7%	<0.0001	
Notes.

PET – Preeclampsia; HTN – Hypertension; NPL – Non progressive labor; PROM – Prelabor rupture of membranes; NRFHR – Non reassuring fetal heart rate; CS – Cesarean section.

Table 2 presents the perinatal characteristics. Patients in the diabetes only group had higher rates of hydramnios (11.9% vs. 3.1% in the rest of the population; p < 0.001), labor dystocia of first and second stage (5.4% vs. 3.3%; p < 0.001), non-reassuring fetal heart rate (2.7% vs. 1.9%; p < 0.001), and placenta previa (0.7% vs. 0.4%; p < 0.001), than the other study groups. Women in the hypothyroidism only group had a higher rate of oligohydramnios (3.1% vs. 2.4%; p = 0.066), prelabor rupture of membranes (PROM) (11.7% vs. 7.8%; p < 0.001), and preterm delivery and PROM (1.9% vs. 1.2%; p = 0.01), than other study groups. Patients with the combination of hypothyroidism and DM had a higher rate of mild and severe preeclampsia (14.0% vs. 4.2%; p = 0.001), chronic hypertension (11.1% vs. 1.2%; p < 0.001), preterm delivery (14.0% vs. 7.4%; p = 0.01), infection in amniotic fluid (1.8% vs. 0.7%; p = 0.099), induction of labor (39.2% vs. 17.5%; p < 0.001), as well as urgent and non-urgent cesarean sections (44.4% vs. 13.3%; p < 0.001).

Table 3 Neonatal characteristics and outcomes.

Characteristic	Healthy
(n = 217,296 deliveries, N = 83,074 women)	Diabetes
(n = 13,324 deliveries, N = 9,771 women)	Hypothyroidism
(n = 1,502 deliveries,
N = 1,100 women)	Hypothyroidism + diabetes
(n = 171 deliveries,
N = 146 women)	P-value	
Male gender	51.2%	52.8%	51.1%	49.7%	0.0048	
SGA	5.8%	3.2%	4.4%	2.3%	<0.0001	
LGA	9.1%	22.3%	8.3%	21.6%	<0.0001	
Weight mean (g)	3174 ± 549	3329 ± 577	3174 ± 577	3249 ± 638	<0.0001	
<1500 g	1.4%	0.8%	1.3%	0.6%		
1500–2500 g	6.8%	5.9%	8.5%	10.5%		
>2500 g	91.9%	93.3%	90.2%	88.9%		
Gestational age mean (weeks)	39.2 ± 2.3	38.7 ± 1.95	38.96 ± 2.3	38.3 ± 2.3	<0.0001	
<28	0.6%	0.2%	0.9%	0.6%		
28–32	0.8%	0.7%	0.9%	0.6%		
32–34	0.8%	1%	0.8%	2.3%		
34–37	5%	7.6%	6%	8.8%		
>37	92.8%	90.6%	91.5%	87.7%		
Apgar 1 min <5	5.4%	5.5%	4.6%	5.8%	0.5219	
Apgar 5 min <7	3.5%	2.5%	3%	1.8%	<0.0001	
Overall fetal mortality	1.4%	0.9%	1.1%	0.6%	0.0002	
APD	0.7%	0.5%	0.9%	0.6%	0.0758	
IPD	0.1%	0%	0.1%	0%	0.3703	
PPD	0.6%	0.3%	0.2%	0%	0.0017	
Shoulder distortion	0.2%	0.5%	0%	0%	<0.0001	
Malformation of nervous system	0.3%	0.3%	0.3%	0%	0.5524	
Notes.

SGA – Small for gestational age; LGA – Large for gestational age; APD – Antepartum death; IPD – intra-partum death; PPD – postpartum death.

Table 3 presents neonatal characteristics and outcome. Women with the combined endocrinopathy had a higher rate of preterm delivery between 32–34 weeks (2.3% vs. 0.8%; p = 0.024), late preterm birth (8.8% vs. 5%; p < 0.024), and a higher rate of newborns with a birthweight below 2500 g (11.1% vs. 8.2%; p = 0.165).

Three GEE models were constructed to determine independent risk factors for cesarean section (Table 4), preterm delivery (Table 5) and preeclampsia (Table 6) after adjustment for confounding factors: (1) Hypothyroidism [OR 1.6; 95% CI 1.52–1.68)], DM [OR 1.74; 95% CI 1.52–1.99)] and their interaction term (dual endocrinopathy) [OR 3.46; 95% CI 2.53–4.75)], were all independent risk factors for cesarean delivery (Table 4). (2) Hypothyroidism, maternal age, infertility treatments, history of preterm birth, infection of amniotic fluid, preterm PROM and chronic hypertension were all independent risk factors for preterm birth. The interaction between DM and hypothyroidism had no significant association with preterm delivery (Table 5). Finally, (3) Hypothyroidism [OR 1.39; 95% CI 1.23–1.57)], the combination of hypothyroidism and DM [OR 1.82; 95% CI 1.16–2.84)], older maternal age and chronic hypertension were all independent risk factors for preeclampsia (Table 6). However, we did not observe an effect modification of DM by hypothyroidism, as the main effect of DM turned to non-significant in the presence of the interaction.

Table 4 Factors associated with cesarean section based on GEE model.

Patients’ characteristic	OR (95% CI)	P-value	
Hypothyroidism alone	1.6 (1.52–1.68)	<0.0001	
Diabetes mellitus alone	1.74 (1.52–1.99)	<0.0001	
Hypothyroidism and Diabetes Mellitus	3.46 (2.53–4.75)	<0.0001	
Age, years	1.1 (1.099–1.106)	<0.0001	
Parity, 2–5 deliveries vs. 1 delivery	0.82 (0.79–0.85)	<0.0001	
Parity, >6 deliveries vs. 1 delivery	0.57 (0.54–0.6)	<0.0001	
Severe PET	5.68 (5.13–6.28)	<0.0001	
Mild PET	1.24 (1.16–1.32)	<0.0001	
NPL stage I	27.1 (25.1–29.3)	<0.0001	
NPL stage II	3.73 (3.42–4.08)	<0.0001	
NRFHR	9.61 (8.91–10.4)	<0.0001	
Malpresentation	26.5 (25–28.1)	<0.0001	
LGA	1.59 (1.52–1.65)	<0.0001	
Male gender	1.09 (1.07–1.12)	<0.0001	
Chronic HTN	1.56 (1.43–1.7)	<0.0001	
Notes.

GEE - Generalized estimating equations; PET – Preeclampsia; NPL – Non progressive labor; NRFHR – Non reassuring fetal heart rate; LGA – Large for gestational age; HTN – Hypertension.

Table 5 Factors associated with preterm delivery based on GEE model.

Patients’ characteristic	OR (95% CI)	P-value	
Hypothyroidism alone	1.14 (0.996–1.3)	<0.0001	
Diabetes mellitus alone	0.86 (0.58–1.28)	0.0567	
Hypothyroidism and Diabetes Mellitus	1.86 (0.87–3.98)	0.4613	
Age, years	1.02 (1.01–1.03)	0.1101	
Parity, 2–5 deliveries vs. 1 delivery	0.42 (0.39–0.46)	<0.0001	
Parity, >6 deliveries vs. 1 delivery	0.4 (0.35–0.46)	<0.0001	
Infertility treatment	1.35 (1.13–1.61)	<0.0001	
History of preterm delivery	1.75 (1.53–1.99)	0.0008	
Infection of amniotic fluid	4.09 (3.37–4.97)	<0.0001	
PPROM	5.37 (4.64–6.2)	<0.0001	
Male gender	0.91 (0.85–0.97)	<0.0001	
Chronic HTN	5.38 (4.66–6.2)	0.0049	
>3 Spontaneous abortion	1.11 (0.91–1.35)	<0.0001	
Notes.

GEE - Generalized estimating equations; PPROM – Preterm prelabor rupture of membranes; HTN – Hypertension.

Table 6 Factors associated with PET based on GEE model.

Patients’ characteristic	OR (95% CI)	P-value	
Hypothyroidism alone	1.39 (1.23–1.57)	<0.0001	
Diabetes mellitus alone	1.04 (0.82–1.31)	0.7631	
Hypothyroidism and Diabetes Mellitus	1.82 (1.16–2.84)	0.0090	
Age, years	1.05 (1.04–1.06)	<0.0001	
Parity	0.46 (0.43–0.48)	<0.0001	
Gestational age	0.9 (0.89–0.91)	<0.0001	
Infertility treatment	0.97 (0.8–1.19)	0.7806	
Chronic HTN	2.57 (1.95–3.39)	<0.0001	
Notes.

GEE – Generalized estimating equations; HTN – Hypertension.

Conducting a similar sensitivity analysis based on different subtypes of diabetes mellitus (gestational diabetes mellitus, diabetes mellitus type I and diabetes mellitus type II) showed similar results, and therefore is not detailed in this paper.

The principal findings of our study show that the combination of hypothyroidism and DM during pregnancy is associated with an increased rate of infertility, hypertensive disorders of pregnancy, preterm deliveries and cesarean sections. Moreover, this dual endocrinological combination was found to be an independent risk factor for cesarean section and for the development of preeclampsia.

Discussion

Diabetes and hypothyroidism are the two most common endocrinopathies during pregnancy. Both conditions have been previously shown to be associated with various pregnancy complications affecting both the mother and the neonate. The association between hypothyroidism and metabolic syndrome including DM and insulin resistance is a subject of extensive studies. Feely & Isles (1979) reported that among diabetic patients 2.7% had also overt hypothyroidism, while the prevalence of subclinical hypothyroidism reached up to 30% in these patients (Feely & Isles, 1979). Other studies reported a prevalence of 10.8–13.4% of thyroid diseases (mostly hypothyroid disorders) in diabetic patients, and the highest rates were recorded among type I diabetes patients and in females (Perros et al., 1995; Smithson, 1998; Gallas et al., 2002; Van sen Driessche et al., 2009; Papazafiropoulou et al., 2010).

In recent years there has been ongoing research exploring the connection between hypothyroidism and insulin resistance (Maratou et al., 2009; Duntas, Orgiazzi & Brabant, 2011). Mannisto et al. (2010) found that overt hypothyroidism during pregnancy increases one’s risk to develop diabetes (OR of 7) later in life (Mannisto et al., 2010). A supportive evidence for this association is the finding that treatment with metformin suppresses TSH secretion (Vigersky, Filmore-Nassar & Glass, 2006; Isidro et al., 2007; Cappelli et al., 2009; Duntas, Orgiazzi & Brabant, 2011). It has been proposed to screen diabetic patients or patients at risk for GDM, for thyroid dysfunction, especially those with DM type I, positive thyroid antibodies, and with TSH concentrations in the upper limits of normal range (Olivieri et al., 2000; Kadiyala, Peter & Okosieme, 2010; Duntas, Orgiazzi & Brabant, 2011).

The study reported herein is the first study to explore the epidemiology of the combined pathology of hypothyroidism and DM during pregnancy. The prevalence of this combined pathology in our population is 0.17% of all women (0.07% of all deliveries). However, the differences in the prevalence of the disease in the different ethnic groups of our region suggest that the prevalence of the disease may be higher and there is under diagnosis of this combined pathology among the rural and nomadic population of our region.

The combination of DM and hypothyroidism during pregnancy, although rare, is associated with a higher rate of hypertensive disorders of pregnancy (See Fig. 2). Indeed, our study showed a higher incidence of hypertensive disorders in the combined endocrinopathy group (25%) than each endocrinopathy by itself and the GEE model demonstrated that the patients with combination of hypothyroidism and DM were at higher risk factor for preeclampsia even after controlling for maternal age, parity and other confounding factors.

Figure 2 The prevalence of hypertensive disorders among the study groups.

Preeclampsia

The association between hypothyroidism as well as DM with the development of preeclampsia is well documented. Indeed, the incidence of preeclampsia among women with hypothyroidism range between 11–44%, especially among those with overt hypothyroidism, although both overt and subclinical hypothyroidism patients showed a higher rate of pregnancy related hypertensive disorders than the general population (Davis, Leveno & Cunningham, 1988; Leung et al., 1993; Poppe & Glinoer, 2003; Wilson et al., 2012; Sahu et al., 2010). All types of hypertensive disorders are more prevalent among patients with DM (type I or II and GDM) (Garner et al., 1990; Leung et al., 1993; Hanson & Persson, 1998; Sibai et al., 2000; Bryson et al., 2003; Yang et al., 2006; Coghill, Hansen & Littman, 2011; Sullivan, Umans & Ratner, 2011). Colatrella et al. reported that the prevalence of chronic hypertension was up to 18%, preeclampsia up to 15% (50% when nephropathy is pre-existing), and gestational hypertension up to 28% in pregnant women with different types of DM (Colatrella et al., 2010). In addition, the risk of developing hypertensive disorders was found to be 1.5 times greater among women with GDM (Bryson et al., 2003), and the risk for eclampsia 1.8 times greater (Coghill, Hansen & Littman, 2011), than other pregnant women. Poor glycemic control as well as the presence of microvascular complications may increase the risk for preeclampsia and its co-morbidities (Hanson & Persson, 1998; Colatrella et al., 2010). A possible underlying mechanism for this observation is the effect of insulin resistance and glucose intolerance on the development of preeclampsia. It was shown that even within the normal range, the level of plasma glucose one hour after a 50 g oral glucose challenge test correlated with the likelihood of preeclampsia. Insulin resistance might play a role in the development of these hypertensive disorders during pregnancy (Joffe et al., 1998; Erez-Weiss et al., 2005; Ness & Sibai, 2006). Nevertheless, in the current study, DM was not an independent risk factor for preeclampsia while the combined endocrinopathy of hypothyroidism and DM was associated with an adjusted odds ratio of 1.82 (95%CI 1.16–2.84) to develop preeclampsia. Notably, these patients had several additional risk factors for preeclampsia, including advanced maternal age (Coghill, Hansen & Littman, 2011; Joseph et al., 2005), a higher rate of infertility treatments (Erez et al., 2006) and chronic hypertension, which may contribute to the inherent risk of the combined endocrinopathy for the development of preeclampsia. These results may suggest that these two endocrinopathies, which modify an effect of one another, and are influenced by metabolic processes, might have an additive influence on the risk to develop hypertensive disorders during pregnancy.

The mechanism through which diabetes and hypothyroidism increase the risk for hypertensive disorders and preeclampsia is complex. Several studies have shown that lipid metabolism disorders, such as hypertriglyceridemia are positively associated with preeclampsia (Ray et al., 2006; Wiznitzer et al., 2009). Both GDM and hypothyroidism are related to elevated plasma triglycerides concentrations (Wiznitzer et al., 2009; Pearce, 2004). Furthermore, pregnancy is a state in which triglycerides and cholesterol levels are elevated (Wiznitzer et al., 2009; Basaran, 2009). Other studies suggest that endothelial dysfunction may also play a role in the development of preeclampsia (Shanklin & Sibai, 1989).

Patients with hypothyroidism show impaired blood flow in response to tissue ischemia or to the administration of endothelial dependent vasodilators, suggesting endothelial dysfunction in these patients (Taddei et al., 2003; Dagre et al., 2007). Thus, the presence of hypothyroidism might enhance the risk for hypertension and preeclampsia.

Cesarean section

The finding that the combined endocrinopathy is an independent risk factor for cesarean section is novel. The majority of the cesarean deliveries were non-urgent (See Fig. 3), this observation may be the result of several contributing factors including: maternal age, high rate of infertility treatments and malpresentation (Bianco et al., 1996; Joseph et al., 2005; Bayrampour & Heaman, 2010). In addition to the indications listed above, hypothyroidism, as well as DM, are associated each with an increased risk for cesarean delivery. Previous studies reported that patients with GDM have an increased rate of cesarean section that may be as high as 23–35% (Casey et al., 1997; Langer et al., 2005; Beucher, Viaris de Lesegno & Dreyfus, 2010). Moreover, Sahu et al. (2010) reported a relatively high prevalence of 44–52% cesarean sections in a small group of patients with hypothyroidism, compared to 36% in the control group (Sahu et al., 2010). Wasserstrum & Anania (1995) reviewed 43 pregnancies of women with hypothyroidism and reported an increased rate of cesarean section due to fetal distress in women with ‘severe hypothyroidism’ (Wasserstrum & Anania, 1995). A previous study based on our population database, found a cesarean sections rate of 30% in pregnancies complicated by hypothyroidism (Cohen et al., 2011). Collectively these reports are in accord with our findings, of an incidence of 23% and 27% cesarean section in the hypothyroidism and DM groups, respectively. Yet, by themselves they cannot serve as an explanation for the high incidence of cesarean delivery observed in the combined endocrinopathy group. This rate reached 44% and aside the possible etiologies, it seems that an additional driving force for this high rate of cesarean sections is the physicians’ decision to operate due to underlying illnesses, advanced maternal age and other pathologies.

Figure 3 The incidence of urgent and non-urgent cesarean sections among the study groups.

Preterm delivery

Both GDM and pregestational diabetes are associated with a higher risk for preterm delivery (Sibai et al., 2000; Yang et al., 2006; Beucher, Viaris de Lesegno & Dreyfus, 2010). Sibai et al. (2000) found increasing rates of preterm birth related with the severity of pregestational diabetes (Sibai et al., 2000). The Hyperglycemia and Adverse Pregnancy Outcome (HAPO) study (2008) evaluating the association between glucose concentrations and different pregnancy outcomes, showed slightly higher risk for preterm delivery in women with elevated glucose concentrations at 1 and 2 h after 75 g oral glucose tolerance test (HAPO study, 2008).

Hypothyroidism is associated with preterm delivery. Casey et al. (2005) conducted a prospective study which found that women with subclinical hypothyroidism were 1.8 times more likely to deliver prior to 34 weeks of gestation (Casey et al., 2005). Stagnaro-Green et al. (2005) showed similar tendency as women with elevated TSH were 3 times more likely to deliver prior to 32 weeks of gestation (Stagnaro-Green et al., 2005). In our study, we also observed that the combined endocrinopathy is associated with a higher rate of preterm birth. Of interest this group had a higher rate of infection in amniotic fluids, hypertensive disorders and PROM, conditions that may lead to induced or spontaneous preterm deliveries. The lack of independent association between the combined endocrinopathy and preterm birth in the GEE model suggest that the higher rate of preterm birth observed in this patients reflects the underlying pregnancy complications that lead to preterm parturition, rather than the effect of the combination of hypothyroidism and DM by itself.

Strength and limitations of the study

The retrospective design and the fact that the diagnoses are based on ICD-9 coding and not by specify the criteria of diagnosis of these endocrinopathies (for example blood test results, presence of autoimmune antibodies etc.) are the limitations of the current study. Moreover, the lack of information regarding how tight the thyroid function and glucose concentrations were under control and the BMI of the patients may also serve as possible confounding factors. Nevertheless, the strength of our study is the large population it includes. There are 87,000 women and more than 232,000 deliveries included in our analysis, and this cohort has sufficient power to characterize such a rare group of patients who have hypothyroidism and DM.

Conclusions

The findings in our study, especially regarding the increased risk for cesarean section and preeclampsia in women with this combined endocrinopathy, emphasizes the importance of managing the pregnancies of these women with extra care, especially since cesarean section and preeclampsia might result in subsequent maternal and neonatal complications. Therefore it might be reasonable and of benefit, to screen women in reproductive age, which have been diagnosed with one of these endocrinopathies for the other, and monitor carefully the blood pressure of these patients.

The paper was presented as poster # S-128 at the Annual Scientific Meeting of the Society of Gynecologic Investigation–March 2012, San Diego CA, USA. It was also presented as a poster in the joint Israeli Society of Maternal Fetal Medicine with the Israeli Society of Ultrasound in Obstetrics and Gynecology on November 2012, Tel Aviv, ISRAEL.

Additional Information and Declarations

Competing Interests

Author Contributions

Human Ethics

Offer Erez is an Academic Editor for PeerJ.

Dan Tirosh and Offer Erez conceived and designed the experiments, analyzed the data, wrote the paper.

Neta Benshalom-Tirosh and Lena Novack analyzed the data, wrote the paper.

Fernanda Press, Ruthy Beer-Weisel, Arnon Wiznitzer and Moshe Mazor wrote the paper.

The following information was supplied relating to ethical approvals (i.e. approving body and any reference numbers):

The study was approved by the Institutional Review Board of the Soroka University Medical Center, Beer Sheva, ISRAEL.

Case number - SOR-0045-11.

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
