# Peer review of "Hypothyroidism and diabetes mellitus – a risky dual gestational endocrinopathy"

_PeerJ, doi:10.7717/peerj.52_

## Round 0.1 · original submission · Minor Revisions

I hope you will seriously consider revising and resubmitting your manuscript.

Reviewer 1 ·

Basic reporting

The paper adheres to the policies.

Experimental design

In the paper Hypothyroidism and Diabetes Mellitus – a Risky Dual Gestational Endocrinopathy, the authors Dan Tirosh and colleagues explore a question whether pregnant women with both hypothyroidism and diabetes experience more adverse obstetric outcomes then the rest of the population. This is a retrospective observational study divided women into the following groups: no endocrinopathies, a single endocrinopathy: diabetes mellitus or hypothyroidism alone vs combined hypothyroidism and diabete based on ICD-9 diagnosis codes. The records were obtained form a computerized database, which included 232 393 deliveries by 87,213 women. The results of this analysis show that patients in the studied population who carried both the diagnosis of hypothyroidism and diabetes mellitus had higher rates of infertility, preeclampsia, chronic hypertension, pre-term birth and cesarean deliveries. Further statistic modeling reveals that odds ratios for preeclampsia and cesarean delivery are higher in women who have both endocrinopathies rather than one, but increased risk of pre-term delivery was not independently associated. The only significant neonatal outcome for the double endocrinopathy group was increased rate of prematurity (32-34 and 34-37 weeks of gestation), with increased percentage of children born with low weight (1500-2000g, but not <1500). Number of children large for gestational age was similar in both DM and DM+hypthyroidism group.

Combined endocrinopathy is a relatively uncommon, but as the obesity epidemic continues and rates of gestational diabetes rise, it may become much more prevalent. To date there is relatively little written on the interplay of the two conditions co-existing in pregnancy, which certainly makes this introductory paper a worthwhile endeavor. Large sample size of the data base is also a strength of this paper. On the other hand, the main limitation is that diagnosis is based on ICD codes, which may not reflect physiologic state of thyroid replacement. Also, there is no information about the degree of glycemic control, which could be an important confounder. As authors have mentioned, it is also likely that this study design could have resulted in under-reporting of hypothyroidism due to under-diagnosis of women from nomadic groups. While it would be a better design to be able to correlate outcomes with TSH and degree of glucose control, it is still an interesting observation that patients having a dual endocrinopathy per history had increased risks above that of each condition alone.

Validity of the findings

The authors need to include a further discuss the limitations of the paper. For example, the lack of information about glycemic control and BMI of the women are potential confounding factors. It can be conceivable that there may be a disproportionate obesity rate in the combined endocrinopathy group, which may play a role in hypertension and affect outcomes. If information on weight or BMI is available, the authors may consider adding it and re-assessing the data. If the information is not available, please discuss potential confounding factors.

Additional comments

Overall the paper is written well. I would suggest that the authors consider splitting the Results and Discussion section to increase the general clarity of organization. I would also like to see p values in the Results section whenever a percentage comparison is made. On the other hand, I had significant trouble with the Tables section because the annotations of P values and what is statistically significant was not clear. For example, the star marking significance in table 1 ( <0.0001) is placed by the title of the table. This would imply to me that every value is significant. I suspect that the authors did not mean to say that there is a statistical significance in the history of fetal mortality between groups (2.4% vs 2.3%). Please revise this. I have similar comments for table 2 and 3.

Additional minor comments:
1. In the Figure section, Figure 1 can be omitted – I do not think it adds much since the section is well explained in the text of the paper.
2. I would like clarification of the prevalence of dual endocrinopathy. It is listed in the abstract as 0.07%. My understanding from the paper is that 0.07% is the percentage of births affected by both conditions. The actual prevalence of both conditions in mothers is 146/87,213, which is 0.17%. Please re-word this for accuracy in the abstract.
3. Women with a dual endocrinopathy appear to have fewer pregnancies (1.17 vs 2.62) in addition to having higher rates of infertility treatment during the study period. If there is a statistically significant difference, you may consider mentioning it in the paper.
4. Information included in Tables 4-6 would work well presented in a graph form (e.g. forest plot), but still having the actual numbers included, as you did in the table.
5. Please enter a reference for the sentence in the introduction: “Fetuses of pregnant women with uncontrolled hypothyroidism are at a greater risk to develop multinodular goiter and have a higher tendency to be small or large for gestational age.”

·

Basic reporting

No comment

Experimental design

Major Comment:

Given the topic, I expected to see an effect modification term (some refer to this as an interaction term) described in the methods. Specifically, the reader will likely be interested in seeing whether the magnitude of association between DM and pregnancy outcome differs significantly by whether hypothyroidism is comorbid. The odds ratios showing these relationships are presented; but the p-value of the effect modification term is not. Thus, the basis for the combination of DM and hypothyroidism is unclear if there is no significant difference in the magnitude of association between one and the pregnancy outcome among strata defined by the other condition. For example, the confidence intervals of the point estimate for the magnitude of association between hypothyroidism and preeclampsia and separately combined hypothyroidism plus DM overlap, suggesting no difference in magnitude comparing isolated versus comorbid hypothyroidism- yet, the manuscript does not provide this level of detail absent a p-value characterizing the effect modification (i.e., interaction) term.

Minor Comment:
1. You may want to refer to the logit link function rather than a binary outcome after introducing the GEE model.
2. It is also preferable that you describe how candidate potentially confounding factors were selected for consideration, and what framework was used to perform model reduction and how/why this was chosen.

Validity of the findings

The authors' findings appear both valid and interesting. Additional detail as requested above would strengthen this manuscript, though.

Additional comments

This is an interesting study and the authors should be commended for their effort.

---

## Round 0.2 · accepted · Accept

Thank you for your submission to PeerJ. I am writing to inform you that your manuscript, "HYPOTHYROIDISM AND DIABETES MELLITUS – A RISKY DUAL GESTATIONAL ENDOCRINOPATHY" (#2012:11:83:1:0:REVIEW), has been accepted for publication.